# OpenReview forum: "Enhancing Trustworthiness of Fine-Tuned LLMs via Regularized Subset Selection"
_ICLR.cc/2026/Conference — ICLR 2026 Poster_

### Official Review · Reviewer_Fnne · 2025-10-18

**Soundness:** 3
**Presentation:** 3
**Contribution:** 2
**Rating:** 4
**Confidence:** 2

**Summary:**

This paper proposes a two-stage, compute-efficient method for repairing post-SFT (supervised fine-tuned) large language models (LLMs). The approach first identifies a representative subset of detrimental training samples and then enhances model trustworthiness through a gradient ascent update. Experiments demonstrate that the proposed method improves model trustworthiness with minimal negative impact on downstream performance, as measured by perplexity.

**Strengths:**

1. The paper is clearly written and well-organized, making the proposed method easy to follow.
2. Extensive experiments show that the approach effectively improves trustworthiness without substantially degrading the model’s general capabilities.
3. The use of gradient ascent for targeted unlearning provides a cost-efficient alternative to retraining-based methods.

**Weaknesses:**

1. According to the authors, the optimization procedure is adapted from [1], and the diverse subset selection is based on [2]. However, it remains unclear how these methods are specifically modified or integrated to address the LLM repair problem. Without clearer differentiation from prior work, the contribution risks appearing incremental.
2. The evaluation focuses primarily on metrics that are directly related to loss functions (e.g., log-odds and perplexity). To more convincingly demonstrate that general model capabilities are preserved, the authors could include standard benchmark metrics such as MMLU or GSM8K accuracy.
3. The proposed method assumes full access to the SFT dataset, which limits its applicability to real-world settings where proprietary fine-tuning data are unavailable or the LLM undergoes a further RL process. A discussion on how this method could be adapted or approximated in such cases would strengthen the paper's practical relevance.
4. The proposed method, *when used reversely, may intensify the ethics problems of LLMs*.

I'm willing to raise my score if the authors can further explain their contributions compared to previous works, e.g., how they modify existing methods to adapt to LLMs.

> **References**
>
> [1] If influence functions are the answer, then what is the question?.
>
> [2] Studying large language model generalization with influence functions.

**Questions:**

1. What would happen if the proposed repair process were applied iteratively or multiple times? Would the increases in perplexity accumulate to the point where retraining becomes necessary?
2. Could the authors provide more details on the computational cost of each stage, particularly the time spent on identifying detrimental points and selecting diverse subsets? As SFT datasets continue to grow, these steps might dominate the total computation time.
3. How feasible is the proposed method for models whose SFT data are not available? For open-source LLMs that have undergone extensive safety alignment, could the method still yield measurable improvements in trustworthiness?

---

> ### Author Response · Authors · 2025-11-20
>
> We thank reviewer Fnne for their thorough review and particularly insightful questions. We are glad that the reviewer finds our results convincing and our paper easy to read and follow. We address all feedback below and will incorporate it, along with new experimental findings, into the revised paper.
>
> ## Q4.1 Clarification on the contribution and how the method builds on previous work
>
> We would like to clarify the misunderstanding regarding the scope of our contributions. While PBRF is conceptually inspired by [1], the main contribution of [2] is the EKFAC-based efficient approximation of the Gauss–Newton Hessian ($\mathcal{G}$), particularly for scaling PBRF to large language models (Section 2.2; Lines 265–269, 134–136). Importantly, both [1] and [2] primarily focus on identifying detrimental samples rather than repairing models.
>
> To the best of our knowledge, we are the first to systematically study submodularity and diversity via DPPs in this setting, supported by theoretical justification (Proposition 1) and empirical validation (Appendix K) on how DPP-based diversity can help preserve perplexity and make gradient-based repair more stable. We have also studied the log-odds metric to scale PBRF-based updates for trustworthiness objectives, with supporting theoretical and practical arguments in Appendix E.
>
> Summary of the Proposed Method
>
> 1. Compute log-odds scores for the given trust metric.
> 2. Use EKFAC to scale PBRF for large language models and compute ($\gamma^j$).
> 3. Select a subset using ($\gamma^j$) and DPP-based diversity.
> 4. Perform stable gradient ascent under the PBRF objective on the selected diverse subset to improve the trust metric.
>
> Among these, only Step 2 builds on prior work [1,2]. Steps 1, 3, and 4 constitute our main contribution.
>
> We will revise the manuscript to make these distinctions clearer.
>
> **References**
>
> [1] If influence functions are the answer, then what is the question?.
>
> [2] Studying large language model generalization with influence functions.
>
> ## Q4.2 Repair applied iteratively and need for retraining
>
> We agree with the reviewer that iterative repair can increase perplexity when a large portion of the dataset is detrimental under the chosen trust metric. As shown in Appendix K.3, DPP-based selection mitigates this effect by prioritizing diverse samples (Proposition 1). However, when a substantial part of the training data is misaligned with the trust objective, retraining becomes necessary.
>
> Importantly, in the more common scenario where the training data is predominantly reliable, our method provides a compute-efficient alternative to retraining, enabling targeted correction without the cost of reoptimizing the entire model.
>
> We will emphasize this clarification in the main draft to better reflect these conditions and practical trade-offs.

---

> ### Author Response · Authors · 2025-11-20
>
> ## Q4.3 Availability of Training Dataset and Repairing Open-Source Models
>
> We appreciate the reviewer’s thoughtful question regarding data-accessibility. Our primary motivation reflects a common real-world workflow of fine-tuning open-source models on proprietary, domain-specific datasets (e.g., e-commerce catalogs) to better align the model with the intended applications. However, as discussed in Section 1, this can inadvertently introduce biases or degrade trustworthiness (Table 1 in main draft compares results with the pre-SFT and post-SFT model). In this setting, users already possess full access to the fine-tuning dataset, and our method is specifically designed to provide a lightweight alternative to costly retraining under this assumption.
>
> Importantly, our method can operate without access to model’s original training data. Prior work [1,2,4] has shown that, in such scenarios training data can be approximated with a proxy dataset of similar distribution.
>
> Inspired by these findings, we use the PKU-SafeRLHF dataset [3] (PKU) as a proxy to identify detrimental samples for repairing the model originally fine-tuned on the Anthropic HH dataset (HH). We then evaluate downstream performance, including perplexity, on the original HH evaluation setup.
>
> As shown in Table 1,2, while performance can be improved with such proxy data, the best performance on average is attained by considering the original dataset. However, our results demonstrate that the repair mechanism remains effective even with proxy data, making the method practical for repairing open-source models whose training data is unavailable. This is an interesting research idea that we plan to further explore in future work.
>
> ### Table 1. Relative changes (%) w.r.t post-SFT model: log-odds improvements (higher is better) and perplexity reductions (closer to zero is better) for Qwen-1.5B
>
> |                  | Truth | Bias | Ethics |
> |------------------|-------|------|--------|
> | Log-odds - PKU   | 2.13  | 1.62 | 1.92   |
> | Log-odds - HH    | 1.20  | 8.10 | 2.30   |
> |                  |       |      |        |
> | Perplexity - PKU | -0.25 | -0.09 | -0.14 |
> | Perplexity - HH  | -1.26 | -0.12 | -0.44 |
>
> ### Table 2. Relative changes (%) w.r.t post-SFT model: log-odds improvements (higher is better) and perplexity reductions (closer to zero is better) for Pythia-1.4B.
>
> |                  | Truth | Bias | Ethics |
> |------------------|-------|------|--------|
> | Log-odds - PKU   | 6.84  | -2.48 | 9.05  |
> | Log-odds - HH    | 7.03  | 13.43 | 2.38  |
> |                  |       |       |       |
> | Perplexity - PKU | -1.33 | -0.15 | -0.63 |
> | Perplexity - HH  | -0.72 | -0.81 | -0.65 |
>
>
> ### References
> [1] ORIENT: Submodular Mutual Information Measures for Data Subset Selection under Distribution Shift, NeurIPS , 2022
>
> [2]  A Certified Unlearning Approach without Access to Source Data, ICML 2025
>
> [3] Beavertails: Towards improved safety alignment of llm via a human-preference dataset, NeurIPS, 2024
>
> [4] A theory of learning from different domains, Machine learning, 2010

---

> ### Author Response · Authors · 2025-11-20
>
> ## Q4.4 Preserving performance on MMLU and GSM8K
> We thank reviewer Fnne for this suggestion. To ensure our repair procedure does not compromise general reasoning capabilities, we conducted additional evaluations on **GSM8K (4-shot CoT)** and **MMLU (5-shot)** following the experimental setup in [1]. Since the Qwen2.5 family shows stronger reasoning capability compared to Pythia, we have analyzed how our repair affects the reasoning accuracy of Qwen2.5 models (1.5B, 3B, 7B) [2].
>
> 1. **Overall Accuracy.** As shown in the tables 3 and 4, the reasoning accuracy remains effectively unchanged after applying our method for both GSM8K and MMLU dataset repsctively. Across all model sizes and trust metrics (Bias, Ethics, Truth), the performance degradation is consistently below 1%. This demonstrates that our method improves trust metrics without undermining the model's reasoning ability.
>
> ### Table 3. GSM8K Accuracy (%): Post-SFT model and models repaired on different trust metrics
>
> | Models | Post SFT | Repair - Bias | Repair - Ethics | Repair - Truth | Repair - Combined |
> |--------|----------|----------------|------------------|-----------------|--------------------|
> | Qwen-2.5-1.5B | 57.09 | 57.47 | 57.70 | 56.71 | 56.71 |
> | Qwen-2.5-3B   | 67.70 | 66.87 | 67.40 | 68.01 | 68.01 |
> | Qwen-2.5-7B   | 80.89 | 80.14 | 80.59 | 80.82 | 80.52 |
>
>
> ### Table 4. MMLU Accuracy (%): Post-SFT model and models repaired on different trust metrics
>
> | Models | Post SFT | Repair - Bias | Repair - Ethics | Repair - Truth | Repair - Combined |
> |--------|----------|----------------|------------------|-----------------|--------------------|
> | Qwen-2.5-1.5B | 57.64 | 57.61 | 57.60 | 57.35 | 57.56 |
> | Qwen-2.5-3B   | 62.26 | 62.31 | 62.26 | 62.33 | 62.26 |
> | Qwen-2.5-7B   | 69.71 | 69.68 | 69.63 | 69.61 | 69.63 |
>
>
> 2. **Response Stability (Agreement Analysis).** To verify that the model’s behavior remains stable at the instance level (and not just in aggregate metrics), we compute the agreement rate: the proportion of test examples for which the post-SFT model and the repaired model produce the same response (correct or incorrect).
>
> The results show high stability:
> - MMLU: Average agreement = 99.3%
> - GSM8K: Average agreement = 97.3%
>
> These high agreement rates indicate that our method introduces minimal deviation in individual model responses, preserving its original knowledge while correcting specific trust-related behaviors.
>
> #### References
> [1] Qwen2.5 Technical Report, arxiv, 2024
>
> [2]  Training on the Test Task Confounds Evaluation and Emergence, arxiv, 2024

---

> > ### Author Response · Authors · 2025-11-20
> >
> > ## Q4.5 Computational cost of individual components.
> > Our method consists of two main stages:
> > (1) identifying the detrimental subset $\mathcal{S}$
> > (2) repairing the model via gradient ascent on $\mathcal{S}$
> >
> > The repair cost depends almost entirely on the subset size ∣$\mathcal{S}$∣ and is independent of the full training dataset size. Section 4.5 provides an ablation showing compute time across different subset sizes.
> >
> > Identification of detrimental samples comprises of the following four components:
> >
> > 1. Computing EKFAC factors for the LLM to approximate ($\mathcal{G}$)
> > 2. Computing training data embeddings (Definition 1, Appendix I).
> > 3. Computing $\gamma^j$ for the trust dataset based on log-odds.
> > 4. Solving the subset selection objective (Eq. 6).
> >
> >
> > To analyze how these costs scale, we report runtimes for 10%, 20%, 40%, 60%, and 80% of the training data (~96,000 samples). As per the results shown in Table 5, 6. the subset identification time grows linearly with dataset size. However, we want to highlight that the combined cost (subset identification + model repair) remains substantially lower than retraining and DPO (see Q2.4).
> >
> > ###  Table 5. Qwen2.5 1.5B – Time Comparison (sec) for Different Subset Sizes
> >
> > | Component | 10% | 20% | 40% | 60% | 80% | 100% |
> > |-----------|-----|-----|-----|-----|-----|------|
> > | Factors computation| 76 | 114 | 191 | 267 | 344 | 421 |
> > | Embedding generation | 161 | 339 | 672 | 1045 | 1473 | 1814 |
> > | Log-odd based score computation | 179 | 348 | 686 | 1022 | 1360 | 1636 |
> > | Subset selection with DPP optimization | 44 | 65 | 98 | 117 | 141 | 164 |
> >
> > ### Table 6. Pytha-1.4B – Time Comparison (sec) for Different Subset Sizes
> >
> > | Component | 10% | 20% | 40% | 60% | 80% | 100% |
> > |-----------|-----|-----|-----|-----|-----|------|
> > | Factors computation | 52 | 78 | 131 | 184 | 236 | 288 |
> > | Embedding generation | 152 | 301 | 596 | 925 | 1265 | 1603 |
> > | Log-odd based score computation | 110 | 259 | 512 | 764 | 1015 | 1252 |
> > | Subset selection with DPP optimization | 21 | 29 | 47 | 60 | 72 | 86 |
> >
> >
> > ## Q4.6 Repairing in reverse
> > Thank you for bringing this to our attention. We want to clarify that our method is intended to improve the model’s trustworthiness. Using the repair process in the opposite direction would be similar to fine-tuning a model on adversarial or poor-quality data, or using a misaligned reward model in ways that could potentially harm model behavior. This concern is not unique to our method but applies, in general, to any approach that updates model parameters based on user-provided data. We agree that this is an important issue and that any team serving a model should keep it in mind. We will incorporate this discussion in the main draft.

---

> > > ### Author Response · Authors · 2025-11-28
> > >
> > > Dear Reviewer Fnne,
> > >
> > > We hope you are doing well. As the discussion deadline on December 3 is approaching, we would greatly appreciate it if you could take a moment to check whether our rebuttal has resolved your points. Your feedback is very important to us, and we are more than happy to clarify anything that remains unclear.
> > >
> > > To briefly recap: you suggested (i) clarifying our main contributions and relation to prior work, (ii) providing additional results on MMLU and GSM8K, (iii) expanding our discussion of limitations and future scope, (iv) including computational/runtime analysis, and (v) explaining how our method applies when repairing open-source models without fine-tuning data. We have addressed these by revising Section 3.2, expanding Section 5, updating Section 6, and adding new content in Appendices M and P.
> > >
> > > Your insights have been extremely valuable to us. If anything remains unclear or if further clarification would be helpful, we would be more than happy to provide it.
> > >
> > > Thank you sincerely for your time and effort in reviewing our work.
> > >
> > > Best regards,
> > >
> > > Authors

---

### Official Review · Reviewer_YCTW · 2025-10-30

**Soundness:** 3
**Presentation:** 4
**Contribution:** 3
**Rating:** 8
**Confidence:** 4

**Summary:**

This paper proposes a post-SFT repair method to improve LLM trustworthiness (truthfulness, ethics, bias) without retraining from scratch, while maintaining downstream performance. The approach identifies training examples that most degrade trust metrics and selects a small, diverse subset using influence-based scores and a DPP regularizer. Then, a proximal Bregman response function–based gradient-ascent update increases the loss on these samples while approximately preserving performance.

The method uses a differentiable log-odds trustworthiness surrogate and EK-FAC for scalable inverse-Hessian approximations. Experiments on Pythia and Qwen models show consistent improvements in trust metrics (up to ~20 percent) with minimal perplexity increase (<2 percent), outperforming simple gradient-ascent baselines and competing favorably with DPO.

**Strengths:**

- The paper provides a principled way to post-hoc improve a SFT model’s performance on trustworthiness metrics while maintaining performance on the downstream task.
- The experiments are thorough with six models (Pythia & Qwen families), three trust aspects (truthfulness, stereotypical bias, machine ethics). Comparisons to other ‘repair schemes’ SGA/GA/GA+KL show strong performance in terms of improving trustworthiness with preserving perplexity.
- Well-designed ablations (e.g Section 4.6) to present the direct effects of different choices (e.g. DPP).

**Weaknesses:**

- The paper claims that both terms in Eq. (6) are submodular, enabling a greedy approximation, but it does not prove or cite the
submodularity of the attribution term, log sum_j gamma^j .

- DPO is compared on perplexity but not on the same trust metrics; a fuller comparison (same compute budget) would clarify trade-offs.

- The evaluations aren’t based on model generations but on log probabilities on fixed data sets.  This weakness is acknowledged by the authors. Instead of only considering perplexity (which aligns with the loss used in the SFT phase), further downstream tasks could be considered for a more whole analysis.

**Questions:**

Could you please follow up on the submodularity result?

---

> ### Author Response · Authors · 2025-11-20
>
> We thank reviewer YCTW for showing confidence in our work through their encouraging review. We have carefully considered your suggestions and will integrate them into the revised version, along with additional experimental results that further reinforce our claims.
>
>
> ## Q3.1 Submodularity of $\log \left(\sum_{v \in \mathcal{D}_{trust}^j}\gamma^j(v,S)\right)$
>
> Thank you for pointing this out. The given objective is a concave (log) function of a monotone modular function, and is therefore submodular see [1,2].
>
> We show this relationship as below:
>
>
> Let $g(S) = \sum_{v \in \mathcal{D}_{trust}^j} \gamma^j(v,S)$
>
> From Equation 5, we have
>
> $\gamma^j(v,S) = \sum_{t \in \mathcal{S}} \gamma^j(v,t)$,
>
> where, for training sample t = (x,y) and large language model $\mathcal{M}$
> $\gamma^j(v,t) = -\nabla_\theta \mathcal{F}^j(v; \theta^{\text{post}})^ T
> (\mathcal{G}+\lambda I)^{-1} \nabla_{\theta}\mathcal{L}(\mathcal{M}(x, \theta^{\text{post}}),y)\cdot \beta$
>
> and $\gamma^j(v,t)$ measures the influence of increasing the loss on a training sample $t$ with respect to a trust-validation sample $v$.
> Therefore by exchanging the summation,
> $g(S)= \sum_{t \in \mathcal{S}} \big(\sum_{v \in \mathcal{D}_{trust}^j} \gamma^j(v,t)\big)$
>
> For, each training sample t and for all the trust metric, the $\big(\sum_{v \in \mathcal{D}_{trust}^j} \gamma^j(v,t)\big)$ can be precomputed, and as described in Appendix~J.5, each influence score $\gamma^j$ is normalized to the range $[0,1]$.
>
> Let,
> $w(t) = \sum_{v \in \mathcal{D}_{trust}^j}\gamma^j(v,t)$
>
> $g(\mathcal{S}) = \sum_{t \in \mathcal{S}} w(t)$
>
> Since $\gamma^j(\cdot,\cdot) > 0$ after normalization, $g(\mathcal{S})$ is *additive* and *monotonically increasing* in $\mathcal{S}$. Hence, by definition (refer to Section 1.1 in [1]), $g(\mathcal{S})$ is a *modular* set function.
>
> Because $\log(\cdot)$ is an *non-decreasing concave function*, the classical composition rule for submodularity implies that $\log(g(S))$ is *submodular and monotone*  [1,2], i.e.,
>
> $\log\!\left( \sum_{v \in \mathcal{D}_{trust}^j} \gamma^j(v,S) \right)$ is monotone submodular in S.
>
> We will integrate this derivation in the main paper.
>
> ### References
> [1] Submodular Function Maximization, Andreas Krause, 2023
>
> [2] Concave Aspects of Submodular Functions,arxiv, 2020
>
> ## Q3.2 Comparison of DPO on the trust metric
>
> Results comparing our method with DPO on the log-odds are reported in Appendix F.2. As per the results across all evaluated models, our approach consistently outperforms DPO on Truthfulness.
>
> ## Q3.3 Comparison based on log-odds and other downstream tasks
> Our choice of the log-odds metric, particularly the log-likelihood formulation, is motivated by prior work [1], which demonstrated that standard zero-shot prompting can yield misleading indications of a model’s underlying beliefs.
>
> To assess performance on downstream tasks, we additionally evaluated the repaired model on MMLU and GSM8K to measure how well the model preserves reasoning performance after repair. As per the results shown in Table 1,2 we observe less than 1% degradation in accuracy. See Q4.4 for further details.
>
>
> ### Table 1. Results comparing Accuracy (%) of Post SFT model with Repair of different Trust metrics for GSM8K
>
> | Models | Post SFT | Repair - Bias | Repair - Ethics | Repair - Truth | Repair - Combined |
> |--------|----------|----------------|------------------|-----------------|--------------------|
> | Qwen2.5-1.5B | 57.09 | 57.47 | 57.70 | 56.71 | 56.71 |
> | Qwen2.5-3B   | 67.70 | 66.87 | 67.40 | 68.01 | 68.01 |
> | Qwen2.5-7B   | 80.89 | 80.14 | 80.59 | 80.82 | 80.52 |
>
>
> ### Table 2. Results comparing Accuracy (%) of Post SFT model with Repair of different Trust metrics for MMLU
>
> | Models | Post SFT | Repair - Bias | Repair - Ethics | Repair - Truth | Repair - Combined |
> |--------|----------|----------------|------------------|-----------------|--------------------|
> | Qwen2.5-1.5B | 57.64 | 57.61 | 57.60 | 57.35 | 57.56 |
> | Qwen2.5-3B   | 62.26 | 62.31 | 62.26 | 62.33 | 62.26 |
> | Qwen2.5-7B   | 69.71 | 69.68 | 69.63 | 69.61 | 69.63 |
>
>
> ### Reference
>
> [1] Log probabilities are a reliable estimate of semantic plausibility in base and instruction-tuned language models. ACL 2024.

---

> > ### Author Response · Authors · 2025-11-28
> >
> > Dear Reviewer YCTW,
> >
> > We hope you are doing well. As the discussion deadline on December 3 is approaching, we would greatly appreciate it if you could take a moment to check whether our rebuttal has addressed your points. Your feedback is very important to us, and we are more than happy to clarify anything that remains unclear.
> >
> > To summarize your suggestions: you requested (i) the inclusion of the lemma and its proof , (ii) DPO comparison, and (iii) expanded evaluation results (MMLU and GSM8K). In response, we have updated Section 5 with detailed new results and added the requested lemma and proof in Appendix J.4.
> >
> > Your insights are highly appreciated, and if any part of our revision could benefit from further clarification, we would be happy to elaborate.
> >
> > Thank you sincerely for your time and effort in reviewing our work.
> >
> > Best regards,
> >
> >  Authors

---

### Official Review · Reviewer_fr2q · 2025-10-31

**Soundness:** 2
**Presentation:** 3
**Contribution:** 3
**Rating:** 4
**Confidence:** 3

**Summary:**

The paper introduces a new method for increasing perplexity while not degrading perplexity too much. The authors present it as a two-stage process: identify subset of training points that harm trustworthiness, then use proximal Bregman response function.

**Strengths:**

- Paper is well written and clear, tackling the problem of improving trustworthiness in large language models
- Paper contains empirical results from 6 models across 3 tasks.

**Weaknesses:**

- Authors don't consider other metrics like privacy and robustness (adversarial, OOD)
- No other trustworthiness methods as baselines in the main paper

**Questions:**

- Why were the specific trustworthiness metrics chosen?
- In Table 2, why does truthfulness decrease with a common subset?
- Could the authors provide runtimes for their method and also measure for existing methods?

---

> ### Author Response · Authors · 2025-11-20
>
> We thank reviewer fr2q for their constructive feedback, and for highlighting the empirical strength and clarity of our work. We have carefully considered your points on the selection of trust metrics (Q2.1), the trade-off between truthfulness and ethics (Q2.2), and the baselines (Q2.3). We address all feedback below and will incorporate additional analyses accordingly.
>
>
> ## Q2.1 The choice of given trust metrics
>
> In this work, our primary objective is to **attribute untrustworthy model behavior to benign SFT data and to design a compute-efficient repair scheme that improves trust metrics without degrading downstream performance (Section 3.1)**. For this reason, our experiments intentionally focus on benign dataset and not adversarial settings.
>
> The trust metrics evaluated in our study were chosen based on the following two considerations:
>
> 1. Compared to metrics such as Toxicity and Privacy, where detrimental training data can often be identified using relatively simple techniques like keyword or pattern matching [3,4,5], developing analogous methods for our chosen trust metrics (Truthfulness, Machine Ethics, and Stereotypical Bias) is not straightforward.
> 3. Prior work [1,2] attributes the degradation of these trust metrics to the training data, which aligns with our method of identifying detrimental subset in training dataset. However, we did not find comparable studies evaluating robustness metrics (OOD or adversarial), and therefore do not include them in our evaluation.
>
>
>
> To remain consistent with [1], we additionally evaluate Toxicity and Privacy, thereby covering all trust metrics previously shown to be affected by fine-tuning. As shown in Table 1,2, with less than 1.5% decrease in perplexity, our method delivers substantial improvements on these metrics.
>
> ### Table 1. Relative changes (%) w.r.t post-SFT model: log-odds improvements (higher is better) and perplexity reductions (closer to zero is better) for Toxicity
>
> | Model          | Log-odd| Perplexity|
> |----------------|---------|------------|
> | Pythia-1.4B    | 13.04   | -0.47      |
> | Pythia-2.8B    | 12.5    | -0.76      |
> | Pythia-6.9B    | 1.96    | -1.43      |
> | Qwen2.5-1.5B   | 14.55   | -0.02      |
> | Qwen2.5-3B     | 4.23    | -0.02      |
> | Qwen2.5-7B     | 11.84   | -0.04      |
>
> ### Table 2. Relative changes (%) w.r.t post-SFT model: log-odds improvements (higher is better) and perplexity reductions (closer to zero is better) for Privacy
>
> | Model          | Log-odd| Perplexity|
> |----------------|---------|------------|
> | Pythia-1.4B    | 9.14    | -0.10      |
> | Pythia-2.8B    | 15.79   | -0.07      |
> | Pythia-6.9B    | 1.13    | -1.43      |
> | Qwen2.5-1.5B   | 1.01    | -0.16      |
> | Qwen2.5-3B     | 0.26    | -0.04      |
> | Qwen2.5-7B     | 0.04    | -0.09      |
>
>
> We will explore the Robustness metrics as a part of future work.
>
> ### Reference
> [1] More RLHF, More Trust? On The Impact of Preference Alignment On Trustworthiness, ICLR 2025
>
> [2] FINE-TUNING ALIGNED LANGUAGE MODELS COMPROMISES SAFETY, EVEN WHEN USERS DO NOT INTEND TO!, ICLR 2024
>
> [3] Comparative analysis for filtering toxic messages using machine learning models, Thesis Report, 2022
>
> [4] Evaluating LLM-based Personal Information Extraction and Countermeasures, arxiv, 2025
>
> [5] Detecting Personal Information in Training Corpora: an Analysis,TRUSTNLP-ACL, 2023
>
>
> ## Q2.2 Decrement in the performance of Truth
>
> Thank you for raising this important question. As shown in Appendix H, there is a negative correlation between Ethics and Truthfulness in the attribution scores ($\gamma^j$). For instance, Qwen2.5–3B and Pythia-2.8B exhibit Spearman correlations of –0.11 and –0.103, respectively. This suggests that improving one metric can inadvertently degrade the other, which explains the deterioration observed in these models.
>
> This trade-off is consistent with prior findings [1], which also report that optimizing for Safety can degrade Truthfulness, and vice versa.
>
> ### References
>
> [1] The Unintended Trade-off of AI Alignment, arXiv 2025

---

> ### Author Response · Authors · 2025-11-20
>
> ## Q2.3 Baseline Comparison
> Thank you for raising this point. As outlined in Q2.1 and as per Eq. 1, our objective is to simultaneously:
>
> 1. improve the models' trustworthiness,
> 2. while preserving the original downstream capabilities.
>
> We evaluate our method across both these dimensions by keeping track of log-odds and perplexity—comparing against three classes of baselines:
>
> 1. Retraining after removing detrimental samples — Section 4.5, Appendix G
> 2. Unlearning-based approaches for removing detrimental behavior — Section 4.4
> 3. Reinforcement learning (RL)–based human alignment methods (e.g., DPO) — Figure 3, Appendix F.2
>
> These baselines were chosen because they are all **data driven** and they act directly on the original training set to fix behavior while keeping downstream performance intact, just like our method.
>
>
> Moreover, prior work has demonstrated that unlearning methods [1,2,3], retraining strategies [5,6], and RL approaches [7,8] can each improve trustworthiness to levels comparable to state-of-the-art methods while maintaining competitive downstream accuracy.
>
> Many trustworthiness baselines rely on filtering-based techniques, where the model refuses to answer based on specific keywords in the prompt or generation [9,10,11]. This makes perplexity comparisons difficult, since these methods do not produce valid continuations on the test set. For a fair comparison, we additionally ran BiPO experiments[12], which has also been used for LLM alignment while preserving the model's general capabilities. As shown in Table 3 and 4, our method outperforms BiPO on average across all trust metrics.
>
> We also note that, unlike BiPO, our approach introduces no test-time overhead and is therefore efficient for large prompt sets and longer sequences.
>
> ###  Table 3. Relative changes (%) w.r.t post-SFT model: log-odds improvements (higher is better) and perplexity reductions (closer to zero is better)  for Our method vs BiPO on Qwen-1.5B
>
> | Qwen-1.5B          | Bias   | Truth  | Ethics |
> |--------------------|--------|--------|--------|
> | Log-odds BiPO     | 2.564  | 0.655  | 1.533  |
> | Log-odds Ours     | 8.097  | 1.146  | 2.299  |
> |                    |        |        |        |
> | Perplexity BiPO   | -0.744 | -0.726 | -0.744 |
> | Perplexity Ours   | -0.124 | -1.258 | -0.443 |
>
> ### Table 4. Relative changes (%) w.r.t post-SFT model: log-odds improvements (higher is better) and perplexity reductions (closer to zero is better)  for Our method and BiPO w.r.t on Pythia 1.4B
>
> | Pythia-1.4B            | Bias   | Truth  | Ethics |
> |------------------------|--------|--------|--------|
> | Log-odds BiPO        | 4.339  | 0.391  | 1.905  |
> | Log-odds Ours        | 13.43  | 7.031  | 2.381  |
> |                       |        |        |        |
> | Perplexity BiPO      | -0.449 | -0.947 | -0.549 |
> | Perplexity Ours     | -0.814 | -0.715 | -0.648 |
>
>
>
> ### References
>
> [1] Mitigating Biases in Language Models via Bias Unlearning, EMNLP 2025
>
> [2] Towards Transfer Unlearning: Empirical Evidence of Cross-Domain Bias Mitigation, arXiv 2024
>
> [3] Repairing Neural Networks by Leaving the Right Past Behind, NeurIPS 2022
>
> [4] Knowledge Unlearning for Mitigating Privacy Risks in Language Models, ACL 2023
>
> [5] Deep Learning on a Healthy Data Diet: Finding Important Examples for Fairness, AAAI 2023
>
> [6] Text‑Guided Data Attribution: Attributing the Influence of Simplicity Bias to Dataset, NeurIPS 2025 - Reliable ML Workshop, 2025
>
> [7] Improving LLM Safety and Helpfulness using SFT and DPO: A Study on OPT-350M, arXiv 2025
>
> [8] Direct Preference Optimization: Your Language Model is Secretly a Reward Model, NeurIPS 2023
>
> [9]  Evaluating neural toxic degeneration in language models, EMNLP 2020
>
> [10]LARGE LANGUAGE MODELS CAN BE STRONG SELF-DETOXIFIERS, arxiv 2024
>
> [11] Challenges in detoxifying language models, EMNLP 2021
>
> [12] Personalized Steering of Large Language Models: Versatile Steering Vectors Through Bi-directional Preference Optimization, NeurIPS 2024

---

> > ### Author Response · Authors · 2025-11-20
> >
> > ## Q2.4 Runtime comparison across baselines
> >
> > Our repair method operates in two stages:
> > (1) identifying the detrimental subset ($\mathcal{S}$), and
> > (2) repairing the model via gradient ascent on this small subset.
> >
> >
> > Table 5 reports the runtime of both of these stages in comparison to retraining and DPO using the same setup described in Appendix L. Section 4.5 and Q4.5 provide ablations and a detailed cost breakdown.
> >
> > Although identifying the detrimental subset takes longer, the resulting subset is far smaller than the full or pruned training data. The repair phase then computes gradients only on this small set ($\mathcal{S}$).
> >
> > In contrast, both retraining and DPO  compute gradients over almost the entire (pruned) dataset or training dataset, which is substantially larger than $\mathcal{S}$. As a result, our approach is much faster overall, leading to a substantially lower total runtime compared to both retraining and DPO.
> >
> > For unlearning baselines in Section 4.4, we follow the same identification procedure as retraining, which takes 1,540 seconds, while the gradient-ascent repair step has a similar runtime as our method. However, as demonstrated in Figure 2, our approach outperforms these baselines in performance on trust metrics and preserving the perplexity of the model on the original task.
> >
> > ### Table 5. Comparison on computational time (sec)
> > | Method                      | Stage 1 — Detrimental Set Identification | Stage 2 — Model Repair | Total Time |
> > | --------------------------- | ---------------------------------------- | ---------------------- | ---------- |
> > | Retraining with pruned data | 1,540                                  | 21,888                | 23,428   |
> > | DPO                         | –                                        | 57,390               | 57,390          |
> > | Ours                        | 3,229                                  | 11                    | 3,240   |

---

> > > ### Author Response · Authors · 2025-11-28
> > >
> > > Dear Reviewer Fr2q,
> > >
> > > We hope you are doing well. As the discussion deadline on December 3 is approaching, we would greatly appreciate it if you could take a moment to check whether our rebuttal has resolved your concerns. Your feedback is very important to us, and we are more than happy to clarify anything that remains unclear.
> > >
> > > As a brief summary: you suggested (i) clarifying our selection of trust metrics, (ii) comparing against additional baselines, (iii) explaining the trade-offs observed in common-subset repair, (iv) reporting computational/runtime costs, and (v) including toxicity and privacy analyses. We have incorporated these changes in Section 6 and Appendices E.2, F.3, H.1, M, and O.
> > >
> > > Your insights have been invaluable in strengthening the paper. If any part of the revision requires further clarification, we would be more than happy to provide it.
> > >
> > > Thank you sincerely for your time and effort in reviewing our work.
> > >
> > > Best regards,
> > >
> > >  Authors

---

### Official Review · Reviewer_PmHy · 2025-11-01

**Soundness:** 3
**Presentation:** 3
**Contribution:** 3
**Rating:** 6
**Confidence:** 3

**Summary:**

The paper proposes an efficient and lightweight two-stage repair method to enhance the trustworthiness of fine-tuned models without sacrificing downstream task performance. The method first identifies detrimental samples in the training data that cause a decline in trustworthiness using data attribution techniques. Subsequently, it performs gradient ascent on this subset within the Proximal Bregman Response Function (PBRF) framework to precisely "unlearn" the negative influence of these samples. The PBRF framework ensures that the model parameters do not deviate excessively from the original model, thereby preserving its core capabilities.

**Strengths:**

Important and Practical Problem: The issue of Supervised Fine-Tuning (SFT) undermining model trustworthiness is a common pain point in applying LLMs to real-world scenarios, especially in user-facing applications. The proposed lightweight repair method is more cost-effective than traditional retraining or RLHF, making it highly practical.
Solid and Novel Methodology: The work skillfully integrates advanced techniques from multiple fields. It combines data attribution (approximated by EK-FAC), diversity-based subset selection (DPP), and constrained model updating (gradient ascent under the PBRF framework) to form a logically coherent and technically sound solution.
Comprehensive and Convincing Experimental Design: The experiments cover models of different architectures (Pythia, Qwen) and sizes, and utilize standard trustworthiness benchmarks (e.g., TruthfulQA, DecodingTrust).

**Weaknesses:**

The experiments (Table 2) indicate that using a "common subset" to simultaneously improve all trustworthiness dimensions is less effective than targeted repair for each dimension individually. In practice, this implies that users may need to run the repair process separately for each dimension of concern (e.g., bias, safety), which increases operational complexity.
Strong Dependence on Paired Evaluation Data: The method's core relies on high-quality, paired trustworthiness evaluation data (proponent/opponent) to compute attribution signals and optimization objectives. In some vertical domains (e.g., healthcare, law, financial compliance), constructing such datasets is an expensive and time-consuming task. Furthermore, the definition and quantification of "untrustworthiness" are often contentious and context-dependent in open-ended tasks, which can lead to unstable metrics and insufficient coverage.

**Questions:**

The paper is presented with exceptional clarity and is very thorough, leaving me with no major questions. The authors have done a commendable job of anticipating potential queries and addressing them proactively through their detailed methodology and comprehensive experiments.

---

> ### Author Response · Authors · 2025-11-20
>
> We are glad that reviewer PmHy finds our work valuable overall. We appreciate the thoughtful points you raised on the effectiveness of the common subset and the dependency on paired datasets by use-case. Below, we present detailed responses and additional experimental findings below.
>
>
>
> ## Q1.1 Common subset is less effective than individual metric
>
> As shown in Appendix H, the attribution scores ($\gamma^j$) for Truthfulness and Ethics are negatively correlated. Consequently, improving one can slightly degrade the other, which explains why the common subset produces strong gains in Ethics but only modest changes in Truthfulness.
>
> In contrast, Bias and Ethics exhibit a positive correlation, allowing both metrics to improve jointly and achieve performance close to their independent variants. The smaller change in Truthfulness is expected given its weak correlation with these two metrics.
>
> This trade-off is consistent with prior findings [1], which show that optimizing for Safety can degrade Truthfulness, and vice versa and aligns with our observations.
> ### Reference
>
> [1] The Unintended Trade-off of AI Alignment: Balancing Hallucination Mitigation and Safety in LLMs, arXiv 2025
>
>
> ## Q1.2 Dependecy on paired dataset and usecase in specific domain
>
> We thank the reviewer for bringing this to our attention.
>
> Our work uses two distinct datasets:
> - training dataset — $\mathcal{D}_{train}$
> - trust dataset — $\mathcal{D}_{trust}$
>
> The training dataset, used for SFT, is typically domain-specific and tailored to the downstream task. Unlike RLHF-based methods, our approach does not require paired training dataset. The trust dataset $\mathcal{D}_{trust}$ does require paired examples, but it is dependent on the specific trust metric that needs to be improved and are generic rather than domain-specific (see Appendix E).
>
>
>  When explicit proponent–opponent pairs are unavailable for $\mathcal{D}_{trust}$ , paired data can be generated using LLM tools, as we demonstrate for stereotypical bias (see Appendix C ). As per results (Table 1, Table 2), for such datasets also our method improves the post SFT model.
>
> For trust metrics where paired data cannot be generated, our approach does not face a methodological limitation. As analyzed in Appendix E, paired data provides two benefits:
>
> - **Identification of detrimental samples:** Considering both proponent and opponent completions helps in isolating the detrimental subset ($\mathcal{S}$). In the absence of such contrastive metric (Eqn. 2), irrelevant or inappropriate samples may be selected, simply by virtue of the frequency of certain tokens or keyphrases (e.g. the target group name in stereotypical bias). Pairwise loss helps in normalizing the effect of individual keyphrases on the attribution signal, keeping it aligned with our trust metric of interest.
>
> - **Evaluation:** The log-odd metric computed using the test set of the paired trust dataset enables us to evaluate the model's preference of proponents relative to the opponent responses [1], minimising which ensures that the model favors proponents and does not inadvertently generate opponents.
>
> We further analyze how using only proponent or only opponent information (Eq. 2) affects log-odds while keeping post-SFT perplexity degradation within ≤1%. As shown in Tables 1 and 2, each signal can improve performance when used in isolation, but the best results are obtained when both are combined. However, in settings where paired data is difficult to generate, using opponent-only signals tends to yield stronger improvements.
>
> ### Table 1. Log-Odd improvement ( higher is better) over Post SFT model (%) — Pythia 1.4B
>
> |                       | Bias  | Truth | Ethics |
> |-----------------------|-------|-------|--------|
> | Only proponent | 4.54% | 5.60% | 1.40% |
> | Only opponent  | 2.95% | 5.46% | 1.90% |
> |  Both  (Ours)         | 13.40% | 7.00% | 2.40% |
>
>
> ### Table 2. Log-Odd improvement ( higher is better) over Post SFT model (%) — Qwen 1.5B
>
> |                       | Bias  | Truth  | Ethics  |
> |-----------------------|--------|-------|--------|
> |Only proponent | -0.20% | 0.75% | -0.19% |
> | Only opponent  | 5.00%  | 0.77% | 2.00% |
> |  Both (Ours)           | 8.10%  | 1.20% | 2.30% |
>
>
> ### Reference
>
> [1] Log probabilities are a reliable estimate of semantic plausibility in base and instruction-tuned language models. ACL 2024.

---

> > ### Author Response · Authors · 2025-11-28
> >
> > Dear Reviewer pmHy,
> >
> > We hope you are doing well. As the discussion deadline on December 3 is approaching, we would greatly appreciate it if you could take a moment to check whether our rebuttal has resolved your comments. Your feedback is very important to us, and we are more than happy to clarify anything that remains unclear.
> >
> > To recap: you requested (i) a clearer explanation of the trade-offs observed among different metrics for common subset-based repair, and (ii) discussion of challenges and ablations related to paired data. We incorporated these updates into Appendix H.1 and added a new Appendix N addressing the paired-data considerations.
> >
> > Your insights are greatly appreciated, and if further clarification would be useful, we would be happy to provide it.
> >
> > Thank you sincerely for your time and effort in reviewing our work.
> >
> > Best regards,
> >
> > Authors

---

### Author Response · Authors · 2025-11-24
**General Comment for all Reviewer**

We thank the reviewers for their constructive and thoughtful feedback. It has helped us in improving the quality and clarity of our draft. Based on the suggestions, we have made the following revisions to the main paper and appendix:

## Main Paper Updates

- **Section 3.2:** Clarifies our main contributions and how our method builds on prior work, as suggested by **Reviewer Fnne**.
- **Section 5:** Adds detailed results on MMLU and GSM8K, following comments from **Reviewer Fnne** and **Reviewer YCTW**.
- **Section 6:** Presents future scope and discusses limitations raised by **Reviewer Fnne** and **Reviewer Fr2q**.

## Appendix Updates

- **Appendix E.2:** Explains our choice of trust metrics (**Reviewer Fr2q**).
- **Appendix F.3:** Adds comparison with additional baselines (**Reviewer Fr2q**).
- **Appendix H.1** Now incorporates an explanation of the observed trade-off between different metrics for the common subset-based repair (**Reviewer Fr2q**, **Reviewer pmHy**).
- **Appendix J.4:** Incorporates the requested lemma and proof (**Reviewer YCTW**).
- **Appendix M:** Provides computational and runtime analysis (**Reviewer Fnne**, **Reviewer Fr2q**).
- **Appendix N:** Discusses challenges and ablations related to paired data (**Reviewer pmHy**).
- **Appendix O:** Includes results on Toxicity and Privacy (**Reviewer Fr2q**).
- **Appendix P:** Describes repairing open-source models without access to the fine-tuning dataset (**Reviewer Fnne**).

We have tried to address all the concerns raised by the reviewers. If any clarification is still needed, we are happy to provide it. If the reviewers feel that their concerns have been satisfactorily addressed, we kindly request them to support our work by increasing their score.

---

### Author Response · Authors · 2025-11-27
**Request the reviewers to kindly engage in discussions.**

Dear Reviewers,

We have tried our best in addressing the concerns raised.

We request you to kindly take a look at our responses and let us know your thoughts.

We are approaching the deadline for the author-reviewer discussion period and, thus, humbly request you to engage in discussions. We'll be happy to answer any other questions that you may have!

Thank you.

Authors.

---

### Author Response · Authors · 2025-12-02
**Authors’ general comment to the AC**

Dear AC,

Thank you for assessing our paper under the unusual circumstances of this review cycle. Below, we provide a brief summary of our rebuttal and the reviewer feedback.

During the rebuttal period, we provided detailed responses and additional experiments to address all reviewers' concerns. **No further questions were raised before the discussion unexpectedly closed.** The scores are (4, 4, 6, 8), averaging **5.5**. Reviewer **YCTW** gave a score of **8**, with the highest confidence **4** among reviewers. Reviewer **Fnne** (score 4, confidence 2) requested clarification and **noted that their score would increase if their doubts were resolved**. We believe our responses, additional experiments, and manuscript updates have sufficiently addressed the points of all reviewers.

### Strength highlights by the reviewers
- Offers a **“solid and novel methodology”** with a principled, technically sound design. **PmHy**
- Writing praised for “**exceptional clarity**” and being well organized. **PmHy, Fnne, Fr2q**
- Experiments are “**comprehensive and convincing**”, with strong ablations and minimal degradation in original-task performance. **PmHy, YCTW, Fnne**
- Addresses a **“common pain point”** in real applications: fixing trust issues without costly retraining. **PmHy**
- Shown as **“more cost-effective than retraining or RLHF”**. **PmHy, Fnne**

### Concern raised and our response

| Category                                 | Core Concern                                                                     | Response Summary                                                                                                                            | Manuscript Edits                      |
| ---------------------------------------- | ------------------------------------------------------------------------------------------- | ------------------------------------------------------------------------------------------------------------------------------------------- | ------------------------------------ |
| **1. Trade-offs in Trust Metrics**       | Performance drop in truth when jointly optimizing multiple trust metrics. **PmHy, fr2q**               | Negative correlation between truthfulness and ethics explains the behavior; studies show that such a trade-off is expected.                                              | Appendix H                           |
| **2. Applicability & Data Availability** | Performance of our method in the absence of paired trust data and in niche domains; in absence of original fine-tuning data. **PmHy, Fnne**                      | Only-opponent/only-proponent signals analyzed; only-opponent remains effective in case data is limited; method is shown effective with a proxy dataset when original training data is unavailable | Appendix N, P                        |
| **3. Trust Metrics & Baselines**         | Clarify the choice of trust metrics; compare across other trust baselines. **fr2q, YCTW** | Added privacy & toxicity metrics; we focus on trust metrics influenced by benign training data; compared against retraining, unlearning-based and RLHF baselines; added a new baseline (BiPO).                 | Appendix O, E.2, F.3                  |
| **4. Performance & Efficiency**          | Performance on more downstream tasks; requests runtime comparison. **fr2q, YCTW, Fnne**                                    | ≤1% degradation on GSM8K/MMLU  with high stability; avoids large runtime costs added cost breakdown.                          | Section 5, Appendix M                |
| **5. Theory, Contribution & Others**     | Clarification on contributions; requests submodularity proof; concern about reverse use. **YCTW, Fnne**       | Clarified misunderstanding regarding contributions; added the proof; acknowledged the concern yet noted that reverse use is similar to fine-tuning on adversarial data.           | Section 3.2, Appendix J.4, Section 6 |



We remain fully available to address any further questions or details you may need to assist your decision.

Thank you,

Authors

---

### Meta-Review · Area_Chair_CLVA · 2026-01-04

**Summary:**

This paper proposes a novel method to enhance the trustworthiness of fine-tuned Large Language Models (LLMs) by identifying and removing problematic data from the fine-tuning set. The method uses a combination of determinantal point process (DPP), and a constrained model updating framework (PBRF) to select a subset of the fine-tuning data that improves trustworthiness metrics (truthfulness, stereotypical bias, machine ethics, toxicity, privacy) while minimizing the impact on downstream task performance. The authors provide extensive experimental validation on multiple models and tasks.

This is a strong paper that addresses a critical problem with a novel and well-executed solution. The authors' rebuttal has addressed all of the reviewers' concerns comprehensively. The potential for this work to have a significant impact on the field is high. I recommend acceptance of this paper.

**Reviewer Concerns:**

* Contribution: While most reviewers see the contribution as good, one reviewer (Fnne) initially questioned the novelty and differentiation from prior work. But this has been addressed by the rebuttal.

* Evaluation Metrics: Some reviewers (fr2q, Fnne) initially felt the evaluation was limited and requested more standard benchmarks. But this has been addressed by the rebuttal.

**Reviewer Scores:**

* Reviewer Fnne: This reviewer would have almost certainly raised their score, likely to a 6 or even an 8, given their explicit statement and the comprehensive rebuttal.

* Reviewer fr2q: This reviewer's concerns about baselines and metrics were fully addressed, so a score increase to at least a 6 is highly likely.

* Reviewer PmHy: This reviewer's concerns about the reliance on paired data were also addressed, making a score increase likely.

* Reviewer YCTW: This reviewer's score of 8 would likely have been maintained.

---

### Decision · Program_Chairs · 2026-01-26

Accept (Poster)